# Obesity-Related Ciliopathies: Focus on Advances of Biomarkers

**DOI:** 10.3390/ijms25158484

**Published:** 2024-08-03

**Authors:** Qianwen Zhang, Yiguo Huang, Shiyang Gao, Yu Ding, Hao Zhang, Guoying Chang, Xiumin Wang

**Affiliations:** 1Department of Endocrinology and Metabolism, Shanghai Children’s Medical Center, School of Medicine, Shanghai Jiao Tong University, Shanghai 200127, China; zhangqianwen@scmc.com.cn (Q.Z.); huangyg29@sjtu.edu.cn (Y.H.); gaoshiyang@scmc.com.cn (S.G.); dingyu@scmc.com.cn (Y.D.); 2Heart Center and Shanghai Institute of Pediatric Congenital Heart Disease, Shanghai Children’s Medical Center, National Children’s Medical Center, School of Medicine, Shanghai Jiao Tong University, Shanghai 200127, China; zhanghao@scmc.com.cn

**Keywords:** obesity, ciliopathy, biomarker, rare disease

## Abstract

Obesity-related ciliopathies, as a group of ciliopathies including Alström Syndrome and Bardet–Biedl Syndrome, exhibit distinct genetic and phenotypic variability. The understanding of these diseases is highly significant for understanding the functions of primary cilia in the human body, particularly regarding the relationship between obesity and primary cilia. The diagnosis of these diseases primarily relies on clinical presentation and genetic testing. However, there is a significant lack of research on biomarkers to elucidate the variability in clinical manifestations, disease progression, prognosis, and treatment responses. Through an extensive literature review, the paper focuses on obesity-related ciliopathies, reviewing the advancements in the field and highlighting the potential roles of biomarkers in the clinical presentation, diagnosis, and prognosis of these diseases.

## 1. Introduction

### 1.1. Primary Cilia and Obesity

Primary cilia, also known as sensory cilia, are highly conserved hair-like organelles [1]. Cilia consist of a microtubule core called the axoneme, which extends from a modified centriole known as the basal body. A variety of receptors and ion channels are embedded within the ciliary membranes, facilitating the detection and transmission of stimuli from the extracellular environment, while also being capable of dispatching signals outward [2].

Obesity prevention and control is a global public health challenge. The 2023 World Obesity Map predicts that the worldwide prevalence of obesity will rise to 24% in 2035, with a total of nearly two billion people, and the obese population is also showing a trend in younger groups [3,4]. 

There is a strong link between primary cilia and obesity. A series of obesity-associated loci identified by genome-wide association studies (GWAS) are confirmed to be related to hypothalamic cilia, including adenylate cyclase 3 (ADCY3) and the melanocortin-4 receptor (MC4R). ADCY3 has been seen the marker of cilia in neurons, and MC4R is found to be specifically located on primary cilia in the hypothalamic paraventricular nucleus of mice [5]. 

Meanwhile, the causative genes associated with morbid obesity in humans were later verified to be associated with primary cilia, suggesting that primary cilia may have an important role in obesity and energy metabolism [6]. 

### 1.2. Obesity-Related Ciliopathies

Ciliopathies are a class of genetic disorders whose etiology is related to ciliary dysfunction [7]. The proteins these genes encode are localized in the cilia–centromere complex, affecting the assembly, maintenance or function of the centrioles or cilia [8,9,10]. Cilia are expressed and play roles in most mature cells of the human body; thus, ciliary defects affect nearly all organs and tissues, leading to complex symptoms. Common clinical manifestations include obesity, kidney abnormalities, vision and hearing impairments, heart defects, and insulin resistance [8,10,11,12]. In total, five ciliopathies are characterized with obesity, including Bardet–Biedl syndrome (BBS, OMIM #209900), Alström syndrome (ALMS, OMIM #203800), Carpenter syndrome (CRPT, OMIM #201000), mental retardation, truncal obesity, retinal dystrophy, and micropenis syndrome (MORMS, OMIM #610156), and morbid obesity and spermatogenic failure (MOSPGF, OMIM #615703) [8,13,14,15,16,17]. Most of these diseases are inherited in an autosomal recessive pattern and the pathogenic mutations usually lead to the loss of the protein or the production of a non-functional or truncated protein. BBS has also been described to have a triallelic pattern of inheritance [18].

ALMS is caused by mutations in the *ALMS1* gene, with an incidence of 1–9 per 1,000,000 [13]. Defect *ALMS1* can lead to multi-organ system damage, with primary features including early childhood obesity, insulin resistance and hyperinsulinemia, retinal cone dystrophy, and hearing impairment [19]. The clinical phenotype, onset time, and severity vary significantly among patients [8,13,20,21,22]. 

BBS is also a ciliopathy affecting multiple organ systems, with the estimated incidence of 1 per 160,000 in northern European populations [14]. Unlike ALMS, it could be caused by mutations in many genes, including *BBS1-22*, *IFT74*, *SCLT1*, *SCAPER*, and *NPHP* genes [14,23]. Major clinical manifestations include retinal dystrophy, polydactyly, early-onset obesity, hypogonadism, intellectual disability, and renal abnormalities [23,24]. Secondary clinical manifestations include congenital heart disease, liver involvement (such as liver fibrosis and liver cysts), endocrine disorders, like diabetes, hypothyroidism, and hypercholesterolemia, enamel hypoplasia, ataxia, delayed speech and growth, craniofacial anomalies, and olfactory abnormalities [23,24]. 

CRPT, with an incidence estimated at 1 per 1 million births [25], is characterized by craniosynostosis, polydactyly, cardiac defects, and obesity [26,27,28]. It can be divided into two types. Carpenter syndrome-1 (CRPT1) is caused by homozygous mutations in the *RAB23* gene [28]. According to the Human Gene Mutation Database (HGMD), only 17 pathogenic *RAB23* variants have been described in patients with CRPT1 [29]. Carpenter syndrome-2 (CRPT2) is caused by mutations in the *MEGF8* gene [15]. 

MORMS is caused by homozygous mutations in the *INPP5E* gene [30]. The primary clinical features of MORMS include intellectual disability, truncal obesity, retinal dystrophy, and a micropenis [16,30]. To date, only one family has been reported with this syndrome [16]. Obesity development is one of the hallmark features of MORMS. The case described by Torkar et al. [31] highlighted prominent phenotypic characteristics, including early-onset and severe obesity, accompanied by the development of metabolic syndrome.

MOSPGF is caused by mutations in the *CEP19* gene [17]. The clinical symptom of this disease is much less complicated compared to other ciliopathies. The primary features of MOSPGF include morbid obesity and spermatogenic failure [17]. 

In addition, Joubert syndrome (JBTS, OMIM #213300), as a ciliopathy, is not associated with obesity generally, but, recently, Sophie et al. [32] found that a patient with JBTS caused by mutations in the *ARL13B* gene exhibited an obesity phenotype, expanding the spectrum of JBTS. 

#### 1.2.1. Localization and Functions in Cilia

These proteins encoded by the genes of obesity-related ciliopathies are located in the different accessory substructures of the primary cilia and have unique functions (Figure 1). ALMS1 and CEP19 are located in the basal body [33,34]. CEP19 participates in the process of triggering the entry of intraflagellar transport (IFT) into the cilium [34]. RAB23 localizes to the basal region of the cilium adjacent to one of the centrioles, promoting cilium formation [35]. Eight of the BBS proteins (BBS1, BBS2, BBS4, BBS5, BBS7, BBS8/TTC8, BBS9, and BBS18/BBIP1) form a transport complex called BBSome [36] while proteins BBS6/MKKS, BBS10, and BBS12 are known as chaperone complexes, facilitating the BBSome assembly [37]. In addition, BBS3/ARL6 is a GTPase regulating the BBSome entry to (and exit from) the cilium [38]. ARL13B is also a GTPase ciliary affecting transmembrane protein localizations and anterograde IFT assembly stability though Sonic Hedgehog (Shh) signaling [39]. Proteins encoded by other genes of BBS are believed to be located on the basal body or region, playing roles on recruitment of BBSome [38]. *INPP5E* encodes a 72 kDa phosphatase localized in the axoneme of primary cilia and plays a crucial role in regulating the PI3K signaling pathway within cilia [40]. Megf8 is now believed to take part in forming a membrane-tethered ubiquitin complex that can fine-tune the strength of Hedgehog (Hh) signaling [41].

#### 1.2.2. Clinical Presentations, Diagnosis, and Treatment

There are similarities as well as differences in the clinical presentations between obesity-related ciliopathies due to the different roles of cilia-related genes in various organs. Table 1 categorizes and lists the confirmed clinical phenotypes of five obesity-related ciliopathies, grouped by systems and organs [8,13,14,15,16,17,19,20,21,22,23,24,25,26,27,28,29,30,38,42,43].

Apart from the above three systems, the clinical manifestations presented in different obesity-related ciliopathy are different. In the musculoskeletal system, polydactyly is typical for BBS [8,14,23,24,38], while spinal abnormalities, such as scoliosis and kyphosis, are common in ALMS [13,19,20,21,22,42]. The symptoms vary a lot in CRPT [15,25,26,27,28,29]. 

Patients with BBS, ALMS, and CRPT exhibit urinary system abnormalities, with specific differences observed among these conditions [8,13,14,15,19,20,21,22,23,24,25,26,27,28,29,38,42,44]. Chronic kidney disease is a dominant feature in BBS and could lead to increased morbidity and premature death [45]. The symptom in other diseases is much less severe in comparison with that in BBS [24,45]. Vesicoureteral reflux and nephritis could be observed in patients with ALMS [20], while patients with CRPT usually have hydronephrosis or pyelonephritis [8,26]. In the cardiovascular system, cardiomyopathy is common in ALMS while being rare in other diseases [8,19]. Patients with CRPT are more likely to have congenital heart defects, such as tetralogy of Fallot, patent ductus arteriosus, and atrial septal defect [8]. Abnormalities in the digestive and respiratory systems are usually in BBS and ALMS. Abnormal liver function is common, while non-alcoholic fatty liver disease shows higher incidence in patients with ALMS and BBS. They also exhibit issues related to recurrent respiratory infections [19,23,24]. No abnormalities in these systems were identified in patients with CRPT and MORMS [8,15,16,25,26,27,28,29,30].

The diagnosis of obesity-related ciliopathies primarily consists of two parts: clinical diagnosis and molecular diagnosis [19,24]. There are specific criterial for diagnosis for patients with ALMS and BBS. For example, the diagnosis of BBS can be confirmed if a patient meets four out of six major clinical symptoms or three major symptoms plus two secondary criteria [46]. The clinical diagnosis is made based on the clinical findings (signs and symptoms), medical history, and family history. Genetic sequencing is a powerful tool to confirm a molecular diagnosis. Other strategies, such as PCR and hybridization-based tests, can also be employed, particularly for cascade testing or in populations where there are recurrent hotspot mutations. The number of patients with CRPT, MORMS, and MOSPGF is relatively small, and research on clinical and molecular diagnosis is limited. However, the diagnostic strategy is similar to that for ALMS and BBS.

Currently, symptomatic treatment remains the primary approach for managing obesity-related ciliopathies [47]. For the obesity seen in ALMS and BBS, the MC4R agonist setmelanotide [48] and GLP-1 receptor agonists (GLP-1 RAs) [49] are promising therapeutic options. Notably, the U.S. FDA approved setmelanotide for the chronic management of weight in adults and pediatric patients aged 6 and older with deficiencies in POMC, LEPR, or PCSK1 in 2020 [50]. In June 2022, the indication for setmelanotide was expanded to include patients with BBS [51]. However, the variability in treatment response among patients requires further research to better understand the underlying mechanisms. These new drug targets also have the potential to serve as biomarkers for monitoring treatment efficacy and prognosis.

#### 1.2.3. Characteristics of Obesity

The characteristics of obesity associated with the five ciliopathies are summarized in Table 2. The Table describes the age of onset, prevalence, BMI, and common comorbidities associated with obesity for each condition [14,16,17,19,24,26,28,30]. The onset age and prevalence of obesity vary among the ciliopathies, with BBS, ALMS, and MOSPGF showing early onset obesity [17,19,24], while MORMS typically manifests obesity when patients grow older (5 to 15 years) [16]. The prevalence of obesity is high in ALMS, MOSPGF (91%), and BBS (89%) [17,19,24]. The features of obesity also differ, with central obesity in BBS and truncal obesity in MORMS, whereas MOSPGF is characterized by morbid obesity with BMI > 40.0 kg/m^2^ [17]. CRPT is notable for high birth weight and obesity present at birth, with a prevalence of 90%, but specific onset range and BMI are not reported [26,28].

#### 1.2.4. Mechanism of Obesity

Obesity in the human body is actually the result of an imbalance between energy intake and energy expenditure (Figure 2). Most cells, including neurons and glial cells throughout the central nervous system, possess primary cilia. The hypothalamus is especially critical for the regulation of energy homeostasis. The agouti-related protein (AgRP) neurons and the pro-opiomelanocortin (POMC) neurons in the arcuate nucleus of the hypothalamic tuberal region positively and negatively regulate feeding behavior, respectively [52]. They are also regulated by the periphery, mainly by insulin secreted by the pancreas, leptin secreted by adipocytes, ghrelin secreted by stomach, and cholecystokinin secreted by small intestine. MC4R neurons in the paraventricular nucleus, as their downstream neurons, directly participate in the regulation of energy metabolism [52]. 

The mechanisms associated with ciliary dysfunction and obesity in these syndromes are complex and only partially understood. It is currently believed that obesity in these diseases is associated through the leptin–melanocortin pathway influencing the hypothalamus [7]. Defect function of cilia in these diseases probably affects signaling between neurons/glial cells in this pathway while the causative mechanisms vary in different diseases. Obesity of BBS is caused by defects in the neurological control of the appetite, although it is unclear whether defective leptin signaling [53] or signaling by anorexigenic GPCR neuropeptide Y family receptors is the primary cause [54]. Similar to the situation in BBS, an increase in energy intake was observed both in human and mice with ALMS [55]. The percentage of ciliated hypothalamic neurons is significantly reduced in *Alms1* mutant mice [56]. Apart from that, a significant reduction in energy expenditure also accounts for the process of obesity, though the specific mechanism is still unknown [55]. The obesity in CRPT may result from damaged Hh signaling or *RAB23* itself potentially regulating adipogenesis [28,29].

## 2. Advance of Biomarkers

Due to rarity of patients with obesity-related ciliopathies, the progress in identifying related biomarkers is based predominantly on ALMS and BBS studies [6]. These two syndromes have been the focus of more extensive research, leading to a better understanding of their underlying mechanisms and potential biomarkers for diagnosis, variability in clinical presentation, progress, and prognosis and treatment.

### 2.1. Diagnosis and Differential Diagnosis

The diagnosis of obesity-related ciliopathies primarily involves two components: clinical diagnosis and molecular diagnosis [19,24]. For ALMS and BBS, clinical diagnosis is mainly based on characteristic clinical features, while molecular diagnosis, through genetic testing, is considered the gold standard for definitive diagnosis [19,24]. With the advancement of gene sequencing technologies, next-generation sequencing (NGS) has become the primary method for diagnosing monogenic diseases, including obesity-related ciliopathies [57]. These diseases are autosomal recessive disorders, meaning that the diagnosis can be confirmed by detecting two pathogenic variants in the associated genes. However, there are clinical scenarios where the diagnosis may be challenging. In some cases, patients may exhibit high clinical suspicion for a ciliopathy, while the result of genetic testing may be negative, or only one pathogenic variant is detected, which is insufficient to confirm a diagnosis [58]. 

Gene functional studies can provide new evidence for the ACMG classification of gene variants. Chen et al. [59] conducted a study on patients with a clinical suspicion of ALMS who carried one pathogenic variant and one variant of uncertain significance (VUS). They performed ALMS1 protein expression staining on skin fibroblasts and discovered that ALMS1 expression at the centrosome was severely impaired. This finding provided evidence to prove the pathogenicity of the missense variant of *ALMS1* and confirmed the diagnosis of ALMS. In contrast, for patients carrying only VUS, staining results indicated that ALMS1 expression was not impaired, excluding the diagnosis of ALMS; however, further genetic testing confirmed the diagnosis of BBS. Thus, this study suggests that ALMS1 expression detection could be a promising differential diagnostic marker for ALMS. 

### 2.2. Variability in Clinical Presentation, Progress, and Prognosis

Obesity-related ciliopathies exhibit highly complex and variable clinical manifestations. As genetic disorders, genotype–phenotype analyses can partly explain the variability in clinical presentations, disease progression, and prognosis. 

#### 2.2.1. Genotype–Phenotype Correlation

The clinical manifestations of obesity-related ciliopathies are highly complex, with the same disease potentially caused by different genes. The clinical presentation of BBS can vary depending on the specific gene involved. In a meta-analysis by Niederlova et al. [38], differences in clinical symptoms caused by various BBS genes were compared (Table 3). Patients with *BBS3* mutations tend to have fewer clinical symptoms. Retinal dystrophy is usually milder in patients with *BBS1* gene mutations while being severe in patients with *BBS2*, *BBS3*, and *BBS4* Mutations. Patients with *BBS2* mutations have a higher incidence of polydactyly, and *BBS10* gene mutations are linked to more pronounced obesity and insulin resistance [38]. 

Unlike BBS, which is associated with many genes, ALMS is a monogenic disease. However, the *ALMS1* gene is very large, and different mutation sites within this gene can result in varying severity, progression, and prognosis of clinical manifestations [58]. Patients with variants in exon 16 usually present with early onset (before one-year old) of retinal degeneration, urinary system dysfunction, dilated cardiomyopathy, and diabetes, while those with variants in exon 8 are prone to have milder symptoms or later onset of kidney disease [60]. A recent meta-analysis conducted by Brais Bea-Mascato et al. [22] revealed important insights into the genotype–phenotype, based on data from 227 ALMS patients. Patients with the longest allele of the *ALMS1* gene truncated around exon 10 (E10) exhibit a higher prevalence of liver dysfunction and experience worse disease progression. However, no significant differences in the prevalence of dilated cardiomyopathy (DCM), hypertrophic cardiomyopathy (HCM), and type 2 diabetes mellitus (T2DM) are observed among patients grouped by longevity of allele. Savas et al. found that the c.7911dupC (p. Asn2638Glnfs*24) mutation can be related to severe cardiomyopathy in ALMS [61].

#### 2.2.2. Other Biomarkers from Multi-Omics Data

Additionally, advances in multi-omics data have also made great contributions to the discovery of new biomarkers (Table 4).

Fatty liver is a common issue in these diseases and can progress to liver fibrosis in advanced stages. This condition is related to obesity but may also be linked to pathogenic genes. ALMS is currently considered a classic model for non-alcoholic fatty liver disease (NAFLD) [62].

For monitoring fatty liver disease, liver biopsy is considered as the gold standard. However, there is a limit of widespread use in clinic considering its invasiveness and complexity. Non-invasive indexes, such as the alanine aminotransferase (ALT)/aspartate aminotransferase (AST) ratio, the AST-to-platelet ratio index (APRI), and the Fibrosis-4 Index (FIB-4) could predict liver fibrosis in liver disease to some extent [63,64]. Recently, imaging methods, such as transient elastography (TE), FibroScan, and shear-wave elastography (SWE), have advanced. Silvia et al. [62] found that SWE plays an important predictive role in the progression of fatty liver to fibrosis in patients with ALMS.

Cardiomyopathy is another significant clinical manifestation of ALMS. It can be identified and monitored using biomarkers, such as NT-proBNP, high-sensitivity troponin, and T-wave inversion on a 12-lead electrocardiogram [65]. However, Nicola et al. [66] suggested that evidence for using NT-proBNP to monitor cardiomyopathy in ALMS may be insufficient. They thought that extracellular volume (ECV) expansion might be a more robust predictor of adverse cardiovascular outcomes. Additionally, they also proposed that elevated triglycerides could be used as a potential marker for cardiac fibrosis, but more prospective studies are needed to validate this.

Agnieszka et al. [67] conducted a study to detect and analyze microRNA (miRNA) expression in the serum of patients with ALMS, BBS, obese controls, and normal controls. They found that miR-301a-3p expression was significantly reduced in both ALMS and BBS patients. Meanwhile, miR-92b-3p expression was decreased in ALMS but increased in BBS. Additionally, they identified eight miRNAs (miR-30a-5p, miR-92b-3p, miR-99a-5p, miR-122-5p, miR-192-5p, miR-193a-5p, miR-199a-3p, and miR-205-5p) that showed significant correlations with clinical parameters, including lipid profiles, serum creatinine, cystatin C, fasting glucose, insulin, C-peptide levels, HbA1c values, and insulin resistance (HOMA-IR). These findings suggest that miRNAs could serve as valuable biomarkers of disease progression in patients with ALMS and BBS syndromes. However, further research is needed to validate these results and to fully understand the potential of miRNAs as biomarkers for these conditions.

Krzysztof et al. [68] conducted a non-targeted metabolomics analysis on patients with ALMS, BBS, obese controls, and normal controls. They found that metabolic changes in ALMS/BBS patients were similar to those observed in non-syndrome obesity, such as the higher levels of acylcarnitines, and tetrahydroaldosterone-3-glucuronide, which have been correlated with insulin resistance and hypertension, respectively, in numerous studies [69,70,71]. They also found that ALMS/BBS patients were characterized by elevated levels of oxidized phosphatidylcholines (PCs), which are recognized as markers of oxidative stress. These diseases exhibit clinical manifestations that often progress with age. Through further age-related analysis, they further found that lipids, primarily lysophosphatidylethanolamines (LPE), are major markers of disease progression. Interestingly, only one metabolite, long-chain fatty acid (FA 26:1; O2), showed a negative correlation with age in ALMS and BBS patients, while it correlated positively in the other groups, suggesting it may have a unique indicative biomarker role [67].

In another study, Krzysztof et al. conducted serum bone metabolism marker testing on patients with ALMS and BBS. They found serum osteocalcin (OC) and urinary deoxypyridinoline (DPD) levels were negatively correlated with the HOMA-IR index. Additionally, serum receptor activator of nuclear factor kappa-Β ligand (s-RANKL) levels were negatively correlated with fasting blood glucose concentrations [72].

Fibrotic changes have been reported in multiple organs in ALMS. Merlin et al. [73] analyzed miRNA expression on peripheral lymphocytes from six ALMS patients and found that fibrosis-related miR-324-5p was upregulated by 2.1-fold in ALMS males. Additionally, passenger strand members of the same mature miRNA category (miR-27a vs. miR-27a star), along with other miRNAs previously reported to be associated with fibrosis (miR-27a, miR-27b, miR-29b, and miR-25), showed disturbances. These findings suggest that miRNAs could potentially serve as biomarkers for fibrosis in ALMS.

Chronic kidney disease (CKD) is the most common cause of death among BBS patients, with significant variability in severity. Early identification and intervention are crucial for managing CKD in these patients [24]. Magnetic resonance diffusion tensor imaging (DTI) can be used to assess the microstructural integrity of the kidneys. Compared to the control group, BBS patients exhibited lower cortical fractional anisotropy (FA) and axial diffusivity, and higher mean diffusivity and radial diffusivity in the kidneys [74]. The urine protein profile of BBS patients can provide information on the risk and predictive factors for adverse renal outcomes and urinary markers of renal insufficiency. Notably, the abundance of urinary fibronectin (u-FN), CD44 antigen, and lysosomal α-glucosidase significantly correlated with glomerular filtration rate [75]. By comparing the urinary metabolomics of BBS patients and control subjects, they discovered that the excretion of several monocarboxylates, including lactate, was increased in the early and late stages of CKD [76]. Emanuela et al. conducted a targeted serum metabolomics study comparing BBS patients to control subjects. They found that renal insufficiency in BBS patients was associated with abnormal levels of plasma phosphatidylcholines and acylcarnitines. Miriam et al. assessed the renal function of 54 BBS patients. They discovered that maximum urine osmolality (max-Uosm) was correlated with the annual decline in estimated glomerular filtration rate (ΔeGFR), suggesting that a defect in urine concentration may predict disease progression [77].

**Table 4 ijms-25-08484-t004:** Other biomarkers from multi-omics data.

Clinical Features	Correlation Detection		Biomarkers/Specific Methods	Comment	References
Fatty liver	Imaging analysis		SWE, TE	SWE plays an important predictive role in the progression of fatty liver to fibrosis in patients with ALMS	Silvia et al. [62]
	Metabolomics		ALT/AST ratio, APRI, FIB-4	Predicts liver fibrosis in liver disease to some extent	Guangqin et al. [63], Erin et al. [64]
Cardiomyopathy	Imaging analysis		ECV	ECV expansion might be a more robust predictor of adverse cardiovascular outcomes	Nicola et al. [66]
	Metabolomics		NT-proBNP, high-sensitivity troponin, elevated triglycerides	Elevated triglycerides are proposed as a potential marker for cardiac fibrosis	Ashwin et al. [65], Nicola et al. [66]
Kidney	Proteomics		u-FN, CD44 antigen, lysosomal α-glucosidase	u-FN, CD44 antigen, and lysosomal α-glucosidase significantly correlated with glomerular filtration rate	Marianna et al. [75]
Fibrosis	MiRNA sequencing		miRNAs (miR-324-5p, miR-27a, miR-27b, miR-29b, and miR-25)	MiRNAs could potentially serve as biomarkers for fibrosis in ALMS	Merlin et al. [73]
Obesity and related metabolic syndromes	MiRNA sequencing		miRNAs (miR-30a-5p, miR-92b-3p, miR-99a-5p, miR-122-5p, miR-192-5p, miR-193a-5p, miR-199a-3p, and miR-205-5p)	MiR-301a-3p expression was significantly reduced in both ALMS and BBS patients; miR-92b-3p expression was decreased in ALMS but increased in BBS; miRNAs could serve as valuable biomarkers of disease progression in patients with ALMS and BBS	Agnieszka et al. [67]
	Metabolomics	Non-targeted metabolomics analysis	PC, LPE, long-chain fatty acid (FA 26:1; O_2_), acylcarnitines, tetrahydroaldosterone-3-glucuronide	LPEs are major markers of disease progression; only long-chain fatty acid (FA 26:1; O_2_) showed a negative correlation with age in ALMS and BBS patients	Krzysztof et al. [68]
		Serum bone metabolism marker testing	OC, DPD, s-RANKL	OC and DPD levels were negatively correlated with the HOMA-IR index	Krzysztof et al. [72]

SWE, shear-wave elastography; TE, transient elastography; ECV, extracellular volume; ALMS, Alström syndrome; BBS, Bardet–Biedl syndrome; ALT, alanine aminotransferase; AST, aspartate aminotransferase; APRI, AST-to-platelet ratio index; FIB-4, Fibrosis-4 Index; miRNA, microRNA; PC, oxidized phosphatidylcholine; LPE, lysophosphatidylethanolamines; OC, osteocalcin; DPD, deoxypyridinoline; HOMA-IR, Homeostasis Model Assessment of Insulin Resistance; s-RANKL, serum receptor activator of nuclear factor kappa-Β ligand; u-FN, urinary fibronectin.

## 3. Discussion

Primary cilia, present in most mature human cells, play crucial roles in various organ systems, and defects in cilia can lead to widespread clinical manifestations. Obesity-related ciliopathies constitute a small part of ciliopathies; however, they provide important clues to the study of the relationship between primary cilia and obesity in humans. 

However, obesity-related ciliopathies are very rare, making the discovery of biomarkers much more difficult than other diseases. Regarding the importance of biomarkers in diagnosis, monitoring, and follow-up of these diseases, there is an urgent need for relevant biomarker studies and relevant reviews. Therefore, this paper focuses on the topic of biomarkers for obesity-related ciliopathies and reviews the advances of biomarkers in the clinical presentation, diagnosis, and prognosis of these diseases.

For a long time, the underlying causes of these diseases remained unknown, leading to unclear or delayed diagnoses for many patients. The development of genetic diagnostic technologies, especially NGS, has provided solutions, enabling accurate diagnosis [78]. Trio exome sequencing would be a great option considering the autosomal recessive inheritance pattern. Proband-only medical exome sequencing is considered as a more cost-effective test, especially in China [79], and WES has also proved to be a useful and cost-effective tool in diagnosing ciliopathy [80]. Some countries prefer using targeted gene panels, which focus on a specific set of genes known to be associated with ciliopathies [81]. However, the high cost of genetic testing is a significant barrier to widespread implementation, particularly in remote and low-income areas. The disparity in access to genetic testing in China may contribute to the uneven geographic distribution of ALMS diagnosed cases, with patients in remote and impoverished regions being less likely to receive a diagnosis [42]. RNA sequencing [82] and multiplex PCR [83] could be an optional diagnostic technique for complicated heterogeneous ciliopathies. For physicians with limited literature information of these disease, identifying pathogenic mutations with WES is a promising tool to provide clues to diagnose, although it could lead to overuse and the waste of resources.

For a subset of patients with unclear diagnoses, protein functional studies offer a pathway to resolve uncertainties, thereby providing additional evidence for the classification of variants. This approach not only improves diagnostic rates but also potentially provides new insights into disease severity assessment. Chen et al. [59] conducted immunofluorescence staining for *ALMS1* protein expression on skin fibroblasts from 16 ALMS patients. They found that patients with significant *ALMS1* expression fluorescence exhibited later onset and milder vision impairments. 

The development of multi-omics technologies and analytical methods, including metabolomics, proteomics, imaging analysis, and miRNA sequencing, provide new possibilities for discovering novel biomarkers in these diseases. However, these advancements are still largely focused on explaining the variability in clinical manifestations. Additionally, the majority of research concentrated on AS and BBS, with limited data available for other obesity-related ciliopathies. There is currently a lack of biomarkers that can effectively indicate treatment efficacy. Up to now, only the genotype–phenotype correlation result is powerful enough to be used in clinical practice. For the biomarkers identified from multi-omics data, more evidence should be provided in the future to confirm the correlation and to validate their effectiveness and accuracy in clinical use.

A significant challenge in discovering biomarkers for obesity-related ciliopathies is the limited number of patients, which restricts the statistical power for identifying genotype–phenotype correlations and other omics biomarkers. For example, due to the limited number of patients, Krzysztof et al. did not separately conduct non-targeted metabolomics analysis for BBS and ALMS, which is a noted limitation [68].

Understanding these diseases is not only beneficial for the diagnosis and treatment of the disease itself, but also for the discovery of the role of cilia-related genes in obesity, which is expected to lead to the discovery of new therapeutic targets for obesity. For instance, *ALMS1* is considered a biomarker for the diagnosis and prognosis of acute myocardial infarction (AMI) [84], with specific variants, such as G/A variant (rs674804) and glutamic acid repeat polymorphism, being markers for early-onset myocardial infarction [85,86]. Furthermore, Edwige et al. [87] identified a crucial protein–protein interaction between ALMS1 and protein kinase C-α (PKCα), which could be a new pharmacological class for treating IR and its numerous comorbidities, such as type 2 diabetes and CVD, in large populations.

## 4. Conclusions

Obesity-related ciliopathy is rare. Identifying the pathogenic gene with genetic sequencing is the main method to diagnose these diseases, making it a dominant biomarker. Genotype–phenotype correlation could also explain some variability in clinical presentation. Biomarkers identified with multi-omics are restricted to ALMS and BBS patients. There is currently a lack of biomarkers that can effectively indicate treatment efficacy.

## 5. Future Directions

More cost-effective and convenient genetic testing is crucial for enhancing the early diagnosis and management of obesity-related ciliopathies.

It is crucial to conduct international multi-center studies, pooling patient resources to carry out comprehensive analyses and particularly conducting lipidomics analysis in ALMS [68] and genotype–phenotype analyses focusing on different pathogenic variants within the same BBS gene.

Building on these efforts, integrating multi-omics data with longitudinal cohort studies is essential. Prospective studies that follow patients over time can provide valuable insights into the progression of diseases, prognosis, and treatment efficacy. By combining genomic, transcriptomic, proteomic, and metabolomic data, researchers can uncover new biomarkers and better understand their roles in disease mechanisms.

## Figures and Tables

**Figure 1 ijms-25-08484-f001:**
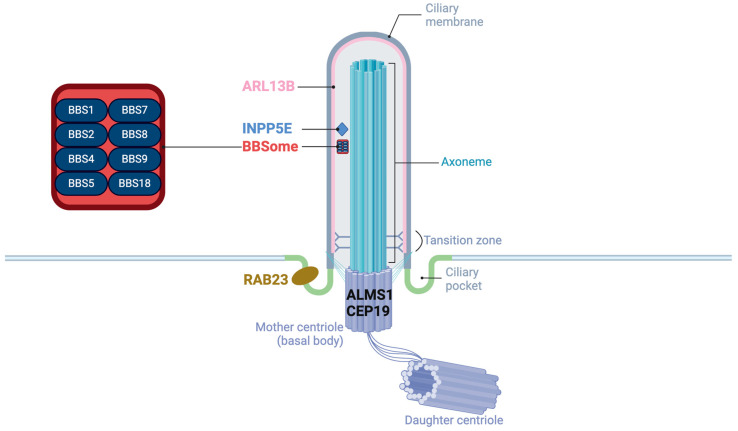
The structure of primary cilia and the localization of proteins of obesity-related ciliopathies.

**Figure 2 ijms-25-08484-f002:**
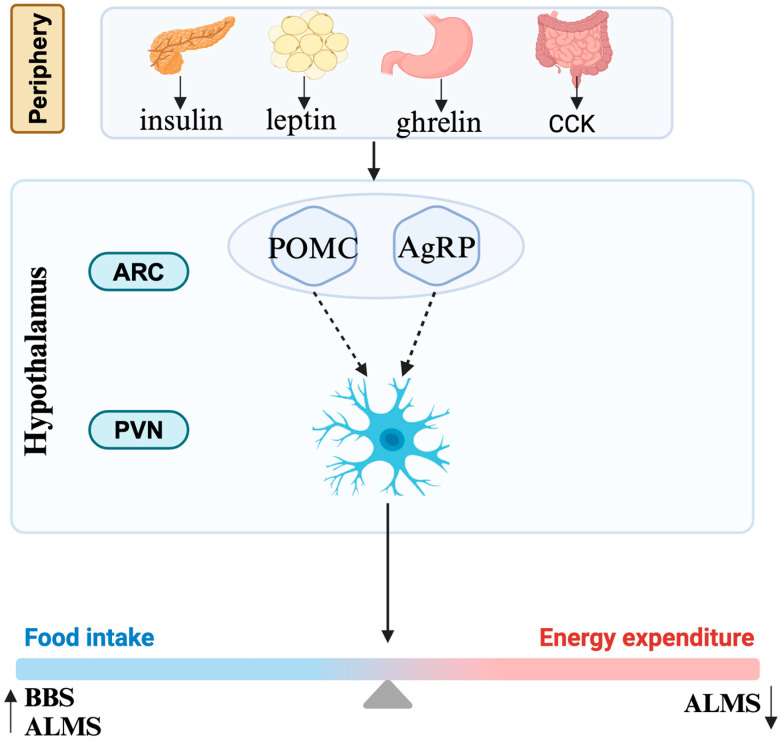
Mechanism of obesity. CCK, cholecystokinin; ARC, arcuate nucleus; PVN, paraventricular nucleus; POMC, pro-opiomelanocortin; AgRP, agouti-related protein; BBS, Bardet–Biedl syndrome; ALMS, Alström syndrome.

**Table 1 ijms-25-08484-t001:** All associated phenotypes of the ciliopathies characterized with obesity.

System or Organ	BBS [8,14,23,24,38,43]	ALMS [8,13,19,20,21,22,42]	CRPT [8,15,25,26,27,28,29]	MORMS [8,16,30]	MOSPGF [8,17]
Nervous system	Macrocephaly, intellectual disability, global developmental delay, cognitive impairment, poor coordination, hydrocephalus, brachycephaly, specific learning disability, neurological speech impairment, gait imbalance, delayed speech and language development, ataxia	Seizures, obsessive-compulsive behaviour, autism, aplasia/hypoplasia of the cerebellum	Trigonocephaly, sagittal craniosynostosis, oxycephaly, large foramen magnum, lambdoidal craniosynostosis, frontal bossing, brachycephaly, intellectual disability, cerebral atrophy, aplasia/hypoplasia of the corpus callosum, coronal craniosynostosis	Delayed speech and language development, intellectual disability, moderate	Intellectual disability
Endocrine system	Obesity, diabetes mellitus, nephrogenic diabetes insipidus, polycystic ovaries, hypertrichosis	Type II diabetes mellitus, insulin-resistant diabetes mellitus, hyperuricemia, hypertriglyceridemia, hyperinsulinemia, gynecomastia, precocious puberty, multinodular goiter, menstrual irregularities, hypothyroidism, hypergonadotropic hypogonadism, growth hormone deficiency, diabetes insipidus, polycystic ovaries, hypertrichosis, alopecia, acanthosis nigricans, global developmental delay, accelerated skeletal maturation, truncal obesity, short stature	Precocious puberty, short stature, obesity	Childhood-onset truncal obesity	Hyperbetalipoproteinemia, hypercholesterolemia, hypertriglyceridemia, insulin resistance, type II diabetes mellitus, obesity
Motion system	Syndactyly, postaxial hand polydactyly, foot polydactyly, mesoaxial polydactyly, finger syndactyly, radial deviation of finger, polydactyly, short foot, broad foot, brachydactyly syndrome	Scoliosis, kyphosis	Joint contracture of the hand, camptodactyly, webbed neck, short neck, toe syndactyly, talipes equinovarus, shallow acetabular fossae, pseudoepiphyses of the proximal phalanges of the hand, preaxial polydactyly, preaxial foot polydactyly, postaxial polydactyly, postaxial hand polydactyly, metatarsus adductus, lateral displacement of patellae, genu varum, genu valgum, duplication of the proximal phalanx of the hallux, cutaneous finger syndactyly, coxa vara, coxa valga, complete duplication of proximal phalanx of the thumb, clinodactyly of the 5th finger, broad thumb, brachydactyly syndrome, aplasia/hypoplasia of the middle phalanges of the toes, aplasia/hypoplasia of the middle phalanges of the hand, aplasia of the middle phalanx of the hand, scoliosis, pectus excavatum, pectus carinatum, flared iliac wings, sacral dimple, spina bifida occulta	NM	NM
Urinary system	Multicystic kidney dysplasia, renal insufficiency, renal hypoplasia, renal cyst, Stage 5 chronic kidney disease, renal agenesis, nephrotic syndrome, abnormality of the kidney, recurrent respiratory infections	Vesicoureteral reflux, tubulointerstitial nephritis, renal insufficiency, abnormality of the urethra, nephrocalcinosis, nephritis, glomerulopathy	Hydroureter, hydronephrosis		
Reproductive system	Cryptorchidism, hypospadias, micropenis, hypoplasia of penis, hypogonadism, vaginal atresia, external genital hypoplasia, decreased testicular size, abnormality of the ovary	Decreased fertility, abnormality of the testis, abnormality of female external genitalia	Shawl scrotum, micropenis, external genital hypoplasia, cryptorchidism	Micropenis	Azoospermia, infertility, oligospermia
Cardiovascular system	Hypertension, left ventricular hypertrophy, bicuspid aortic valve, atria septal defect	Renovascular hypertension, pulmonary hypertension, hypertrophic cardiomyopathy, dilated cardiomyopathy, congestive heart failure, atherosclerosis	Ventricular septal defect, transposition of the great arteries, tetralogy of fallot, patent ductus arteriosus, atria septal defect, pulmonic stenosis	NM	Congestive heart failure, hypertension, myocardial infarction, premature coronary artery disease
Digestive system	Hepatic fibrosis, hepatic failure, biliary tract abnormality, aganglionic megacolon	Splenomegaly, portal hypertension, hepatomegaly, hepatic steatosis, elevated hepatic transaminases, cirrhosis, chronic hepatic failure, chronic active hepatitis	NM	NM	Hepatic steatosis
Respiratory system	Respiratory distress, bronchiolitis, recurrent respiratory infections, asthma	Recurrent pneumonia, asthma, respiratory insufficiency, pulmonary fibrosis	NM	NM	NM
Face	High palate, low-set, posteriorly rotated ears, prominent nasal bridge, short neck, downslanted palpebral fissures, medial flaring of the eyebrow, hypodontia, dental crowding, abnormalities of the teeth	Round face, hyperostosis frontalis interna, gingivitis, abnormalities of the teeth	Wide nasal bridge, upslanted palpebral fissure, underdeveloped supraorbital ridges, sparse eyebrow, retrognathia, preauricular pit, persistence of primary teeth, narrow palate, micrognathia, malar flattening, hypoplasia of the maxilla, hypoplasia of midface, highly arched eyebrow, high palate, depressed nasal bridge, anteverted nares, agenesis of permanent teeth	NM	NM
Eyes	Strabismus, retinopathy, rod-cone dystrophy, abnormal electroretinogram, cataract, myopia, retinal degeneration, cone/cone–rod dystrophy, abnormality of retinal pigmentation, nyctalopia, macular dystrophy, glaucoma, congenital primary aphakia, astigmatism, nystagmus	Deeply set eye, subcapsular cataract, pigmentary retinopathy, photophobia, nystagmus, cone/cone–rod dystrophy, chorioretinal abnormality, blindness	Hypertelorism, epicanthus, telecanthus, optic atrophy, opacification of the corneal stroma, microcornea	Cataract, retinal dystrophy, visual impairment	NM
Ear	Hearing impairment, recurrent otitis media	Progressive sensorineural hearing impairment, otitis media	Sensorineural hearing impairment, protruding ear, low-set ears, conductive hearing impairment, abnormality of the pinna	NM	NM
Other	Hyposmia, situs inversus totalis	Pes planus, hypoalphalipoproteinemia	Umbilical hernia, omphalocele, polysplenia, wide intermamillary distance, supernumerary nipple, hypoplastic nipples, situs inversus totalis, cutis laxa	NM	Hypoalphalipoproteinemia

BBS, Bardet–Biedl syndrome; ALMS, Alström syndrome; CRPT, Carpenter syndrome; MORMS, mental retardation, truncal obesity, retinal dystrophy, and micropenis syndrome; MOSPGF, morbid obesity and spermatogenic failure; NM, not mentioned. From Table 1, it can be deduced that the five genetic ciliopathies exhibit similar phenotypes of the neurologic, endocrine and reproductive systems. In the neurological system, common abnormalities include developmental delays, cognitive impairments, seizures, and motor coordination disorders. In the endocrine system, obesity, diabetes, and insulin resistance could usually be observed. Patients with obesity-related ciliopathies also complain about problems with their reproductive systems, such as genital development issues (e.g., micropenis, vaginal atresia, underdeveloped external genitalia) and reduced fertility (e.g., azoospermia, infertility) [8,13,14,15,16,17,19,20,21,22,23,24,25,26,27,28,29,30,38,42,43].

**Table 2 ijms-25-08484-t002:** Obesity features between BBS, ALMS, CRPT, MORMS, and MOSPGF.

Diseases	Gene	Age of Onset Range	Incidence	BMI	Common Co-Morbidities	Comment
BBS [14,24]	*BBS1-22*, *IFT74*, *SCLT1*, *SCAPER*, *NPHP*	Birth—3 years	89%	35.7 ± 8.0 kg/m^2^	Retinal cone–rod dystrophy, postaxial polydactyly, cognitive impairment, hypogonadism and genitourinary abnormalities, kidney disease	Central obesity; birth weight typically normal.
ALMS [19]	*ALMS1*	Birth—5 years	70–98%	NM	Cone–rod dystrophy, progressive sensorineural hearing loss, short stature, hypogonadism (central or primary), progressive renal disease, insulin resistance/type 2 diabetes mellitus	Birth weight typically normal; hyperphagia and excessive weight gain begin during the first years, resulting in childhood obesity.
CRPT [26,28]	*MEGF8*, *RAB23*	NM	90%	NM	Craniosynostosis, polydactyly, cardiac defects	High birth weight and obesity were prevalent.
MORMS [16,30]	*INPP5E*	5–15 years	NM	NM	Impaired intellectual development, retinal dystrophy, micropenis	Truncal obesity
MOSPGF [17]	*CEP19*	Birth—3 years	91%	>40.0 kg/m^2^	Spermatogenic failure, hypertension, type 2 diabetes mellitus	Morbid obesity

BBS, Bardet–Biedl syndrome; ALMS, Alström syndrome; CRPT, Carpenter syndrome; MORMS, mental retardation, truncal obesity, retinal dystrophy, and micropenis syndrome; MOSPGF, morbid obesity and spermatogenic failure; BMI, body mass index; NM, not mentioned.

**Table 3 ijms-25-08484-t003:** Genotype–phenotype correlation in ALMS and BBS.

Genotype	Distinguishing Clinical Features/Comments
*ALMS1* variants in exon 16 [60]	Early onset (before one year old) of retinal degeneration, urinary system dysfunction, dilated cardiomyopathy, and diabetes
*ALMS1* variants in exon 8 [60]	Milder symptoms or later onset of kidney disease
*ALMS1* truncated around exon 10 [22]	A higher prevalence of liver dysfunction and experience worse disease progression
*ALMS1* c.7911dupC [61]	Severe cardiomyopathy
*BBS1* [38]	Relatively less “syndromic” penetrance of renal anomalies
*BBS2* [38]	Relatively more “syndromic” penetrance of renal anomalies “Leanest” of obesity phenotype
*BBS3* [38] (*ARL6*)	Lowest “syndromic” penetrance of cognitive impairment and renal anomalies
*BBS4* [38]	Low penetrance of renal anomalies Early-onset morbid obesity
*BBS5* [38]	Relatively more “syndromic”
*BBS6* [38] *(MKKS)*	More likely to have CHD and genitourinary malformations
*BBS7* [38]	Relatively more “syndromic” penetrance of renal anomalies
*BBS8* [38] (*TTC8*)	Relatively less “syndromic” penetrance of renal anomalies
*BBS9* [38]	High penetrance of renal anomalies
*BBS10* [38]	Most severe renal impairment Significant adiposity
*BBS12* [38]	Significant adiposity
*BBS21* [38] (*C8rorf37*)	High penetrance of polydactyly

## Data Availability

Not applicable.

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
