# Peer review of "Obesity-Related Ciliopathies: Focus on Advances of Biomarkers"

_ijms, 2024, doi:10.3390/ijms25158484_

Round 1

Reviewer 1 Report

Comments and Suggestions for Authors

This is a narrative review of obesity-related ciliopathies, focusing on biomarkers. The authors have described the different monogenic obesity-related ciliopathies, describing the clinical presentation and genes involved, as well as metabolomic and miRNA studies. In general the review is well written and the presentation well organized and comprehensively described. 

Some comments:

1. In the introduction, only the incidence of ALMS is provided. Perhaps the authors could also include some description on the prevalence of all the ciliopathies described.

2. In line 55, the authors mention that all the ciliopathies they describe are inherited in an autosomal recessive pattern. Perhaps they could further elaborate that this pattern of inheritance involves inheriting one copy of a defective gene from the mother and another copy from the father. 

BBS has also been described to have a triallelic pattern of inheritance. Perhaps the authors could expand on this.

3. In line 103, the abbreviation IFT (intraflagellar transport) is used without any first introduction. Many abbreviations are used throughout the text. It would be useful to have a list of abbreviations. 

4. In table 1 and lines 121-142, it would be good if the specific presentations could be referenced to the appropriate papers in the text.

5. In line 150, the authors claim that genetic sequencing is the only option to confirm a molecular diagnosis. This may be too broad a statement. Other strategies such as PCR and hybridization based tests may also be used particularly for cascade testing or in populations where there are recurrent hotspot mutations.

6. On line 155-156, the MC4R agonist setmelanotide injection has been approved by the US FDA for weight management for BBS patients.

7. On line 351, the authors suggest proband only exome sequencing for molecular diagnosis. However as these ciliopathies are autosomal recessive, trio exome sequencing would be the better option.

Comments on the Quality of English Language

Some sentences need some work on the English, e.g., 

Lines 125 and 142: it would be better to describe patients presenting with certain symptoms rather than complaining about

Line 146: specific criteria for not criterial of

Author Response

  1. In the introduction, only the incidence of ALMS is provided. Perhaps the authors could also include some description on the prevalence of all the ciliopathies described.

Response:  Thank you for your suggestion, reviewer. We have made enhancements to the section regarding the incidence of BBS in lines 65-66 and CRPT in line74. Regarding the reports on MORMS and MOSPGF, the findings are primarily based on a single family study, and there is no available data on the prevalence at this time.

  1. In line 55, the authors mention that all the ciliopathies they describe are inherited in an autosomal recessive pattern. Perhaps they could further elaborate that this pattern of inheritance involves inheriting one copy of a defective gene from the mother and another copy from the father. BBS has also been described to have a triallelic pattern of inheritance. Perhaps the authors could expand on this.

Response: Thanks for the reviewer’s suggestion. Autosomal recessive means that an individual must process two copies of a mutated gene, one from each parent, in order to manifest the disease and BBS has also been described to have a triallelic pattern of inheritance. We have incorporated this explanation into our manuscript in lines 55-59 to provide readers with a clearer understanding of the genetic basis of these conditions. Thank you for helping us improve the clarity of our paper.

  1. In line 103, the abbreviation IFT (intraflagellar transport) is used without any first introduction. Many abbreviations are used throughout the text. It would be useful to have a list of abbreviations.

Response: Thanks for the reviewer’s valuable feedback. We have added a list of abbreviations at the end of the article in lines 437-458 to facilitate better readability for our readers. Thank you for assisting us in enhancing the clarity of our paper.

  1. In table 1 and lines 121-142, it would be good if the specific presentations could be referenced to the appropriate papers in the text.

Response: We apologized for some missing references. We have checked the manuscript and added appropriate citations. We have already made revisions to the content in table 1 and lines 121-146.

  1. In line 150, the authors claim that genetic sequencing is the only option to confirm a molecular diagnosis. This may be too broad a statement. Other strategies such as PCR and hybridization based tests may also be used particularly for cascade testing or in populations where there are recurrent hotspot mutations.

Response: Thanks for the reviewer’s insightful comment regarding line 150. Our statement may be overly broad and we have revised the sentence to provide a more nuanced perspective. Thank you for helping us improve the clarity of our paper.

  1. On line 155-156, the MC4R agonist setmelanotide injection has been approved by the US FDA for weight management for BBS patients.

Response: Thanks for the reviewer’s suggestion. We have further expanded the content regarding the FDA's approval of setmelanotide for use in patients with BBS in lines 161-164.

  1. On line 351, the authors suggest proband only exome sequencing for molecular diagnosis. However as these ciliopathies are autosomal recessive, trio exome sequencing would be the better option.

Response: Thanks for the reviewer’s suggestion. We agreed that trio exome sequencing is a better option but the proband only exome sequencing plus sanger sequencing of parents to confirm the inherence would be more cost-effective. We further discussed it in lines 374-376.

Some sentences need some work on the English, e.g.,

Lines 125 and 142: it would be better to describe patients presenting with certain symptoms rather than complaining about

Response: Thanks for the reviewer’s suggestion, we agree with the suggestion and we have revised the relevant wording accordingly. Thank you for helping us improve the clarity of our paper.

Line 146: specific criteria for not criterial of

Response: Thanks for the reviewer’s suggestion, we have revised the relevant wording accordingly. Thank you for helping us improve the clarity of our paper.

Reviewer 2 Report

Comments and Suggestions for Authors

This review tries to associate ciliopathies with obesity, while identifying phenotypes, genotypes and biomarkers, basically trying to understand the role of cilia-related genes in obesity, that could lead to new therapeutic targets for obesity. However, the structure of the review is not capable of giving that insight. Major changes to improve structure and readability are required, and the following remarks need to be incorporated:

1). Many statements throughout the manuscript are unreferenced.

2). Tables 1,2,3 do not include citations, nor listings of the relevant studies. Exception is Table 4 that is done more scientifically correct. It is suggested to prepare better versions of tables 1,2,3.

3). Several informations are stated twice in different positions of the manuscript. This should be avoided and improved. E.g., Lines 39/43 ‘Meanwhile, … energy metabolism’ is suggested to be repositioned, and merged with the first paragraph of section 1.2; Lines 161/170 repeats in detail the information given on each of the ciliopathies under section 1.2, nevertheless it is REPEATED, and even here (lines 161/170 ) a whole Table is devoted to the obesity association overview. It is suggested to remove the details from section 1.2; The same goes for the genes associated with the ciliopathies, these are mentioned in section 1.2 and then repeated later in the manuscript. The repetition of information is the basic reason why the manuscript seems unfocussed. Making a better structure, and having sections consecutively dealing with one topic will improve the readability.

4). Section 1.2.1 with the description, lines 100-113 with anatomical keywords like ‘cilium’; ‘transport complex’; ‘chaperone complex’; ‘entry to (and exit from) the cilium’; ‘transmembrane protein localizazions’; ‘basal body or region’; ‘axoneme of primary cilia’, could use a Figure that is already introduced during lines 25/29.

5). Paragraph 1.2.4 might be the most essential explanation for the causes of the obesity-related ciliopathies giving rise to disturbed energy homeostasis, but is too short and too unspecific to make a lasting effect. Introducing a figure, with properly explaining and expanding the neurobiology would be essential.

6). Many sentences need editing. It is suggested to proofread the whole manuscript.

Other remarks, of some repeating the above in more detail, are given below:

1.1

Line 26: replace ‘It’ with ‘Cilia’

Lines 27/33: rephrase: ‘A variety of receptors and ion channels are located in the membranes of the cilia, enabling sense and transmitting stimuli from the extracellular environment, while also being able to sent signals to the extracellular environment.’

Line 37: remove ‘et al.,’

Lines 40/41: are OMIM references needed? Isn’t it better to use proper citations?

Line 42: remove ‘later’

Lines 30/33: should be repositioned, a the third paragraph of this section.

Lines 39/43 ‘Meanwhile, … energy metabolism’ is suggested to be repositioned, and merged with the first paragraph of section 1.2; Lines 161/170 repeat a bit more in detail the information given on each of the ciliopathies under section 1.2, nevertheless it is REPEATED, and even here a whole Table is devoted to the obesity association overview. It is suggested to remove the details from section 1.2;

1.2

Line 45: change to ‘Ciliopathies is a class of genetic disorders whose etiology is related to ciliary dysfunction [6].

Line 52 write ‘including’

Lines 55/61: give references for ‘These diseases … hearing impairment.’

Lines 67/73: give references for ‘Major clinical … manifestation of BBS.’

Lines 77/78: give references for ‘Carpenter syndrome … RAB23 gene.’

1.2.1

Basal body

The description, lines 100-113 with anatomical keywords like ‘cilium’; ‘transport complex’; ‘chaperone complex’; ‘entry to (and exit from) the cilium’; ‘transmembrane protein localizazions’; ‘basal body or region’; ‘axoneme of primary cilia’, could use a Figure that is already introduced during lines 25/29.

Lines 103/107: give references for ‘RAB23 localizes … BBSome assembly.’

1.2.2

The references used in line 120 should be positioned and indicated in Table 1.

Line 121: Instead of ‘Five diseases’ use ‘From table 1, it can be deducted that the five genetic ciliopathies exhibit similar phenotypes of the neurologic, endocrine and reproductive systems.’

Lines 121/153: should be referenced accordingly. There is only one reference mentioned (38) in line 149.

Line 128/129: should be rephrased as mentioning ‘other systems’ ‘different obesity-related’ ‘are different’ in one sentence is not clear!

1.2.3

Line 161: instead of ‘of obesity in five’ use ‘of obesity associated with the five

Line 164: use ‘among the’ instead of ‘these’

Lines 161/170 repeat a bit more in detail the information given on each of the ciliopathies under section 1.2, nevertheless REPEAT, and even here a whole Table is devoted to the obesity association overview. It is suggested to remove the identical details from section 1.2

1.2.4

Line 179 write ‘associated through the leptin-melanocortin pathway influencing the hypothalamus’

Please rephrase and use citations: ‘While many cells, including neurons and glial cells throughout the central nervous system, possess primary cilia, it is suggested that the obesity-features associated with the ciliopathies and their manifesting diseases are mainly associated with the Leptin–Melanocortin pathway in the hypothalamus [6]. Especially, considering  that the  hypothalamus is regulating energy homeostasis that is critically affected.

This paragraph might be the most essential explanation for the causes of the obesity-related ciliopathies giving rise to disturbed energy homeostasis, but is too short and too unspecific to make a lasting effect. Introducing a figure, properly explaining and expanding the neurobiology would be essential.

2.

Lines 196/197: should have references and rephrased: ‘Due to rarity in incidence of patients with obesity-related ciliopathies, the progress in identifying related biomarkers is based predominantly on ALMS and BBS studies.’

2.1

Lines 202/209 needs references.

Lines 220/223: add punctuation and more specific writing: ‘In contrast, for patients carrying only VUS, staining results indicated that ALMS1 expression was not impaired, excluding the diagnosis of ALMS, however, frther genetic testing confirmed the diagnosis of BBS. Thus, this study suggests that ALMS1 expression detection could be a promising differential diagnostic marker for ALMS.

2.2.1

Table 3 should be referenced, especially since the corresponding text is referring to different (meta-analysis) studies.

2.2.2

Line 262: rephrase: ‘However, there is a limit to the widespread use of liver biopsy in the clinic considering….’

Line 301: perhaps ‘indicative biomarker role’?

The writing in Section 2.2.2. in combination with Table 4 is the right example of how a review should handle and display the results. References are dealt with in the text as well as in the Table. A remark to the writing is that ‘They found/discovered/etc.’ is used too many times to the point it is disturbing, and it is suggested that the authors have a proper adjustment in academic writing performed.

3

Lines 342/347. The authors claim that knowledge about obesity-related ciliopathies and understanding the role of cilia-related genes in obesity, could lead to new therapeutic targets for obesity. That might be true, but it is not yet obvious from the manuscript. It is suggested to expand on this line of reasoning. The fact that the ciliopathies provide significant insights for common disease research as mentioned in line 380, should be used in this section.

4

Line 289: perhaps: ‘Biomarkers identified with multi-omics are restricted to ALMS and BBS patients.’

Comments on the Quality of English Language

6). Many sentences need editing. It is suggested to proofread the whole manuscript.

Other remarks, of some repeating the above in more detail, are given below:

1.1

Line 26: replace ‘It’ with ‘Cilia’

Lines 27/33: rephrase: ‘A variety of receptors and ion channels are located in the membranes of the cilia, enabling sense and transmitting stimuli from the extracellular environment, while also being able to sent signals to the extracellular environment.’

Line 37: remove ‘et al.,’

Lines 40/41: are OMIM references needed? Isn’t it better to use proper citations?

Line 42: remove ‘later’

Lines 30/33: should be repositioned, a the third paragraph of this section.

Lines 39/43 ‘Meanwhile, … energy metabolism’ is suggested to be repositioned, and merged with the first paragraph of section 1.2; Lines 161/170 repeat a bit more in detail the information given on each of the ciliopathies under section 1.2, nevertheless it is REPEATED, and even here a whole Table is devoted to the obesity association overview. It is suggested to remove the details from section 1.2;

1.2

Line 45: change to ‘Ciliopathies is a class of genetic disorders whose etiology is related to ciliary dysfunction [6].

Line 52 write ‘including’

Lines 55/61: give references for ‘These diseases … hearing impairment.’

Lines 67/73: give references for ‘Major clinical … manifestation of BBS.’

Lines 77/78: give references for ‘Carpenter syndrome … RAB23 gene.’

1.2.1

Basal body

The description, lines 100-113 with anatomical keywords like ‘cilium’; ‘transport complex’; ‘chaperone complex’; ‘entry to (and exit from) the cilium’; ‘transmembrane protein localizazions’; ‘basal body or region’; ‘axoneme of primary cilia’, could use a Figure that is already introduced during lines 25/29.

Lines 103/107: give references for ‘RAB23 localizes … BBSome assembly.’

1.2.2

The references used in line 120 should be positioned and indicated in Table 1.

Line 121: Instead of ‘Five diseases’ use ‘From table 1, it can be deducted that the five genetic ciliopathies exhibit similar phenotypes of the neurologic, endocrine and reproductive systems.’

Lines 121/153: should be referenced accordingly. There is only one reference mentioned (38) in line 149.

Line 128/129: should be rephrased as mentioning ‘other systems’ ‘different obesity-related’ ‘are different’ in one sentence is not clear!

1.2.3

Line 161: instead of ‘of obesity in five’ use ‘of obesity associated with the five

Line 164: use ‘among the’ instead of ‘these’

Lines 161/170 repeat a bit more in detail the information given on each of the ciliopathies under section 1.2, nevertheless REPEAT, and even here a whole Table is devoted to the obesity association overview. It is suggested to remove the identical details from section 1.2

1.2.4

Line 179 write ‘associated through the leptin-melanocortin pathway influencing the hypothalamus’

Please rephrase and use citations: ‘While many cells, including neurons and glial cells throughout the central nervous system, possess primary cilia, it is suggested that the obesity-features associated with the ciliopathies and their manifesting diseases are mainly associated with the Leptin–Melanocortin pathway in the hypothalamus [6]. Especially, considering  that the  hypothalamus is regulating energy homeostasis that is critically affected.

This paragraph might be the most essential explanation for the causes of the obesity-related ciliopathies giving rise to disturbed energy homeostasis, but is too short and too unspecific to make a lasting effect. Introducing a figure, properly explaining and expanding the neurobiology would be essential.

2.

Lines 196/197: should have references and rephrased: ‘Due to rarity in incidence of patients with obesity-related ciliopathies, the progress in identifying related biomarkers is based predominantly on ALMS and BBS studies.’

2.1

Lines 202/209 needs references.

Lines 220/223: add punctuation and more specific writing: ‘In contrast, for patients carrying only VUS, staining results indicated that ALMS1 expression was not impaired, excluding the diagnosis of ALMS, however, frther genetic testing confirmed the diagnosis of BBS. Thus, this study suggests that ALMS1 expression detection could be a promising differential diagnostic marker for ALMS.

2.2.1

Table 3 should be referenced, especially since the corresponding text is referring to different (meta-analysis) studies.

2.2.2

Line 262: rephrase: ‘However, there is a limit to the widespread use of liver biopsy in the clinic considering….’

Line 301: perhaps ‘indicative biomarker role’?

The writing in Section 2.2.2. in combination with Table 4 is the right example of how a review should handle and display the results. References are dealt with in the text as well as in the Table. A remark to the writing is that ‘They found/discovered/etc.’ is used too many times to the point it is disturbing, and it is suggested that the authors have a proper adjustment in academic writing performed.

3

Lines 342/347. The authors claim that knowledge about obesity-related ciliopathies and understanding the role of cilia-related genes in obesity, could lead to new therapeutic targets for obesity. That might be true, but it is not yet obvious from the manuscript. It is suggested to expand on this line of reasoning. The fact that the ciliopathies provide significant insights for common disease research as mentioned in line 380, should be used in this section.

4

Line 289: perhaps: ‘Biomarkers identified with multi-omics are restricted to ALMS and BBS patients.’

Author Response

1). Many statements throughout the manuscript are unreferenced.

Response: We apologized for some missing references. We have conducted a comprehensive review of the manuscript and added appropriate citations.

2). Tables 1,2,3 do not include citations, nor listings of the relevant studies. Exception is Table 4 that is done more scientifically correct. It is suggested to prepare better versions of tables 1,2,3.

Response: Thanks for the reviewer’s suggestion. We have added citations in Tables 1,2,3 to make them more scientifically correct.

3). Several informations are stated twice in different positions of the manuscript. This should be avoided and improved. E.g., Lines 39/43 ‘Meanwhile, … energy metabolism’ is suggested to be repositioned, and merged with the first paragraph of section 1.2; Lines 161/170 repeats in detail the information given on each of the ciliopathies under section 1.2, nevertheless it is REPEATED, and even here (lines 161/170 ) a whole Table is devoted to the obesity association overview. It is suggested to remove the details from section 1.2; The same goes for the genes associated with the ciliopathies, these are mentioned in section 1.2 and then repeated later in the manuscript. The repetition of information is the basic reason why the manuscript seems unfocussed. Making a better structure, and having sections consecutively dealing with one topic will improve the readability.

Response: Thanks for the reviewer’s suggestion. We are sorry for the repetition of the information. We have modified the repeated part ‘Meanwhile, … energy metabolism’ and merged it with the first paragraph of section 1.2. We have deleted the repeated details in section 1.2 which is mentioned later in the manuscript especially for the description of obesity. We also modified the sentence describe the genes associated with the ciliopathies in the manuscript to avoid repetition.

4). Section 1.2.1 with the description, lines 100-113 with anatomical keywords like ‘cilium’; ‘transport complex’; ‘chaperone complex’; ‘entry to (and exit from) the cilium’; ‘transmembrane protein localizazions’; ‘basal body or region’; ‘axoneme of primary cilia’, could use a Figure that is already introduced during lines 25/29.

Response: Thanks for the reviewer’s suggestion. We have added Figure1 to help introduce the structure and the localization of proteins of obesity-related ciliopathies.

5). Paragraph 1.2.4 might be the most essential explanation for the causes of the obesity-related ciliopathies giving rise to disturbed energy homeostasis, but is too short and too unspecific to make a lasting effect. Introducing a figure, with properly explaining and expanding the neurobiology would be essential.

Response: Thanks for the reviewer’s suggestion. We have expanded the explanation of the causes of the obesity-related ciliopathies in lines 186-200 and added a Figure to make it clearer and easier to understand.

6). Many sentences need editing. It is suggested to proofread the whole manuscript.

Response: Thanks for the reviewer’s suggestion.  We appreciate your feedback and agree that a thorough proofreading is essential to ensure the clarity and quality of our work. We have conducted a comprehensive review of the manuscript. We have carefully proofread the text, corrected any grammatical errors, and improved the sentence structure to enhance readability. We have also made sure that the language is consistent and that the manuscript adheres to the standard conventions of scientific writing.

Other remarks, of some repeating the above in more detail, are given below:

1.1

Line 26: replace ‘It’ with ‘Cilia’

Response: Thanks for the reviewer’s suggestion, we have revised the relevant wording accordingly. Thank you for helping us improve the clarity of our paper.

Lines 27/33: rephrase: ‘A variety of receptors and ion channels are located in the membranes of the cilia, enabling sense and transmitting stimuli from the extracellular environment, while also being able to sent signals to the extracellular environment.’

Response: Thanks for the reviewer’s suggestion, we have revised the relevant sentence as "A variety of receptors and ion channels are embedded within the ciliary membranes, facilitating the detection and transmission of stimuli from the extracellular environment, while also capable of dispatching signals outward." Thank you for helping us improve the clarity of our paper.

Line 37: remove ‘et al.,’

Response: Thanks for the reviewer’s suggestion, we have revised the relevant wording accordingly. Thank you for helping us improve the clarity of our paper.

Lines 40/41: are OMIM references needed? Isn’t it better to use proper citations?

Response: Thanks for the reviewer’s suggestion. We have adjusted the content in lines 52-55 accordingly. We believe that OMIM references are necessary, and we have also added relevant citations.

Line 42: remove ‘later’

Response: Thanks for the reviewer’s suggestion, we have revised the relevant wording accordingly. Thank you for helping us improve the clarity of our paper.

Lines 30/33: should be repositioned, a the third paragraph of this section.

Response: Thanks for the reviewer’s suggestion, we have revised the relevant wording accordingly. Thank you for helping us improve the clarity of our paper.

Lines 39/43 ‘Meanwhile, … energy metabolism’ is suggested to be repositioned, and merged with the first paragraph of section 1.2; Lines 161/170 repeat a bit more in detail the information given on each of the ciliopathies under section 1.2, nevertheless it is REPEATED, and even here a whole Table is devoted to the obesity association overview. It is suggested to remove the details from section 1.2;

Response: Thanks for the reviewer’s suggestion. We are sorry for the repetition of the information. We have modified the repeated part ‘Meanwhile, … energy metabolism’ and merged it with the first paragraph of section 1.2. We have deleted the repeated details in section 1.2 which is mentioned later in the manuscript especially for the description of obesity. We also modified the sentence describe the genes associated with the ciliopathies in the manuscript to avoid repetition.

1.2

Line 45: change to ‘Ciliopathies is a class of genetic disorders whose etiology is related to ciliary dysfunction [6].’

Response: Thanks for the reviewer’s suggestion, we have revised the relevant wording accordingly. Thank you for helping us improve the clarity of our paper.

Line 52 write ‘including’

Response: Thanks for the reviewer’s suggestion, we have revised the relevant wording accordingly. Thank you for helping us improve the clarity of our paper.

Lines 55/61: give references for ‘These diseases … hearing impairment.’

Response: We apologized for some missing references. We have checked the manuscript and added appropriate citations.

Lines 67/73: give references for ‘Major clinical … manifestation of BBS.’

Response: We apologized for some missing references. We have checked the manuscript and added appropriate citations.

Lines 77/78: give references for ‘Carpenter syndrome … RAB23 gene.’

Response: We apologized for some missing references. We have checked the manuscript and added appropriate citations.

1.2.1

Basal body

The description, lines 100-113 with anatomical keywords like ‘cilium’; ‘transport complex’; ‘chaperone complex’; ‘entry to (and exit from) the cilium’; ‘transmembrane protein localizazions’; ‘basal body or region’; ‘axoneme of primary cilia’, could use a Figure that is already introduced during lines 25/29.

Response: Thanks for the reviewer’s suggestion. We have added Figure 1 to help introduce the structure and the localization of proteins of obesity-related ciliopathies.

Lines 103/107: give references for ‘RAB23 localizes … BBSome assembly.’

Response: We apologized for some missing references. We have checked the manuscript and added appropriate citations.

1.2.2

The references used in line 120 should be positioned and indicated in Table 1.

Response: We apologized for some missing references in Table 1. We have checked the manuscript and added appropriate citations. We have already made revisions to the content in Table 1.

Line 121: Instead of ‘Five diseases’ use ‘From table 1, it can be deducted that the five genetic ciliopathies exhibit similar phenotypes of the neurologic, endocrine and reproductive systems.’

Response: Thanks for the reviewer’s suggestion, we have revised the relevant wording accordingly. Thank you for helping us improve the clarity of our paper.

Lines 121/153: should be referenced accordingly. There is only one reference mentioned (38) in line 149.

Response: We apologized for some missing references. We have checked the manuscript and added appropriate citations. We have already made revisions to the content in lines 121-146.

Line 128/129: should be rephrased as mentioning ‘other systems’ ‘different obesity-related’ ‘are different’ in one sentence is not clear!

Response: Thanks for the reviewer’s suggestion, we have revised the relevant wording accordingly. Thank you for helping us improve the clarity of our paper.

1.2.3

Line 161: instead of ‘of obesity in five’ use ‘of obesity associated with the five’

Response: Thanks for the reviewer’s suggestion, we have revised the relevant wording accordingly. Thank you for helping us improve the clarity of our paper.

Line 164: use ‘among the’ instead of ‘these’

Response: Thanks for the reviewer’s suggestion, we have revised the relevant wording accordingly. Thank you for helping us improve the clarity of our paper.

Lines 161/170 repeat a bit more in detail the information given on each of the ciliopathies under section 1.2, nevertheless REPEAT, and even here a whole Table is devoted to the obesity association overview. It is suggested to remove the identical details from section 1.2

Response: Thanks for the reviewer’s suggestion. We are sorry for the repetition of the information. We have deleted the repeated details in section 1.2 which is mentioned later in the manuscript especially for the description of obesity. We also modified the sentence describe the genes associated with the ciliopathies in the manuscript to avoid repetition.

1.2.4

Line 179 write ‘associated through the leptin-melanocortin pathway influencing the hypothalamus’

Response: Thanks for the reviewer’s suggestion, we have revised the relevant wording in lines 197-199 accordingly. Thank you for helping us improve the clarity of our paper.

Please rephrase and use citations: ‘While many cells, including neurons and glial cells throughout the central nervous system, possess primary cilia, it is suggested that the obesity-features associated with the ciliopathies and their manifesting diseases are mainly associated with the Leptin–Melanocortin pathway in the hypothalamus [6]. Especially, considering  that the  hypothalamus is regulating energy homeostasis that is critically affected. This paragraph might be the most essential explanation for the causes of the obesity-related ciliopathies giving rise to disturbed energy homeostasis, but is too short and too unspecific to make a lasting effect. Introducing a figure, properly explaining and expanding the neurobiology would be essential.

Response: Thanks for the reviewer’s suggestion, we have revised the relevant wording in 1.2.4 (Lines 185-214) accordingly. Besides, we have added Figure 2 to help introduce mechanism of obesity. Thank you for helping us improve the clarity of our paper.

2.

Lines 196/197: should have references and rephrased: ‘Due to rarity in incidence of patients with obesity-related ciliopathies, the progress in identifying related biomarkers is based predominantly on ALMS and BBS studies.’

Response: Thanks for the reviewer’s suggestion. We have revised the relevant wording in lines 216-217 accordingly. have checked the manuscript and added appropriate citations.Thank you for helping us improve the clarity of our paper.

2.1

Lines 202/209 needs references.

Response: We apologized for some missing references. We have checked the manuscript and added appropriate citations.

Lines 220/223: add punctuation and more specific writing: ‘In contrast, for patients carrying only VUS, staining results indicated that ALMS1 expression was not impaired, excluding the diagnosis of ALMS, however, frther genetic testing confirmed the diagnosis of BBS. Thus, this study suggests that ALMS1 expression detection could be a promising differential diagnostic marker for ALMS.’

Response: Thanks for the reviewer’s suggestion, we have revised the relevant wording in lines 240-244 accordingly. Thank you for helping us improve the clarity of our paper.

2.2.1

Table 3 should be referenced, especially since the corresponding text is referring to different (meta-analysis) studies.

Response: We apologized for some missing references. We have checked the manuscript and added appropriate citations.

2.2.2

Line 262: rephrase: ‘However, there is a limit to the widespread use of liver biopsy in the clinic considering….’

Response: Thanks for the reviewer’s suggestion, we have revised the relevant wording accordingly. Thank you for helping us improve the clarity of our paper.

Line 301: perhaps ‘indicative biomarker role’?

Response: Thanks for the reviewer’s suggestion, we have revised the relevant wording in line 321 accordingly. Thank you for helping us improve the clarity of our paper.

The writing in Section 2.2.2. in combination with Table 4 is the right example of how a review should handle and display the results. References are dealt with in the text as well as in the Table. A remark to the writing is that ‘They found/discovered/etc.’ is used too many times to the point it is disturbing, and it is suggested that the authors have a proper adjustment in academic writing performed.

Response: Thanks for the reviewer’s suggestion, we have revised the relevant wording accordingly. Thank you for helping us improve the clarity of our paper.

3

Lines 342/347. The authors claim that knowledge about obesity-related ciliopathies and understanding the role of cilia-related genes in obesity, could lead to new therapeutic targets for obesity. That might be true, but it is not yet obvious from the manuscript. It is suggested to expand on this line of reasoning. The fact that the ciliopathies provide significant insights for common disease research as mentioned in line 380, should be used in this section.

Response: Thanks for the reviewer’s suggestion. We have merged these two part and expanded it in the last of the discussion.

4

Line 289: perhaps: ‘Biomarkers identified with multi-omics are restricted to ALMS and BBS patients.’

Response: Thanks for the reviewer’s suggestion, we have revised the relevant wording in lines 424-425 accordingly. Thank you for helping us improve the clarity of our paper.

Reviewer 3 Report

Comments and Suggestions for Authors

In general, review articles on rare diseases are often difficult for evaluation since there have been fragment information to describe the figure of diseases. The paper is challenging but includes some interesting descriptions.

1.       Please stress the novelty and/or strength of the paper.

2.       Please describe more the methods and/or policy of selection of the cited reference.

3.       Introduction; the population-based estimated incidence of various diseases (BBS, CRPT, MORMS, MOSPGF) might be added.

4.       Reference numbers appeared to be mismatched in several parts (for instance. Table 4). The overall citations could be checked by the authors.

5.       Please discuss in more detail the approaches for physicians to find the ciliopathies under little literature information. How can the physicians use the biomarkers in the clinics?

6.       Row 25; the first sentence may have a cited reference.

7.       Row 37; the space was necessary between Receptor and (MC4R). In the aspect of space, the overall text could be checked by the authors.

8.       Row 193; resulte or result?

Comments on the Quality of English Language

Minor editing of English language required.

Author Response

  1. Please stress the novelty and/or strength of the paper.

Response: Thanks for the reviewer’s suggestion. We have further stressed the noverty and strength of the paper in the beginning of the discussion part.

  1. Please describe more the methods and/or policy of selection of the cited reference.

Response: Thanks for the reviewer’s suggestion. The method is not a necessary part of the literature review for this journal. We admit that it will be more scientific to describe it but we could not find appropriate place to mention the method part in the manuscript. We will appreciate it very much if you have some more suggestions on this.

Here is the methods:

In this article, we performed a comprehensice literature review on the the advancement of obesity-related ciliopathies and in paticular focuse on the biomarkers. We conducted a mutiple database search (Pubmed, Web of Science, Google) for English language papers published using the following key words: ”ciliopathies”, “obesity”, “primary cilia”, ” Alström Syndrome”, “Bardet-Biedl Syndrome”, “Carpenter syndrome”, and “biomarker”. We also check reference lists of identified articles to find further articles.

  1. Introduction; the population-based estimated incidence of various diseases (BBS, CRPT, MORMS, MOSPGF) might be added.

Response: Thank you for your suggestion, reviewer. We have made enhancements to the section regarding the incidence of BBS in lines 65-66 and CRPT in line74. Regarding the reports on MORMS and MOSPGF, the findings are primarily based on a single family study, and there is no available data on the prevalence at this time.

  1. Reference numbers appeared to be mismatched in several parts (for instance. Table 4). The overall citations could be checked by the authors.

Response: We apologized for some mistakes in some references. We have checked the manuscript and corrected appropriate citations.

  1. Please discuss in more detail the approaches for physicians to find the ciliopathies under little literature information. How can the physicians use the biomarkers in the clinics?

Response: Thanks for the reviewer’s suggestion. We have further discussed about the approaches for physicians who have limited literature information of these disease to identify the ciliopathies in Lines 384-387, and the use of biomarkers in the clinic in Lines 401-404. Unfortunately, most of the biomarkers could not be used in the clinic right now but it is expected to be useful in the future. More evidence should be provided in the future study to confirm the correlation and validate effectiveness and accuracy in clinical use.

  1. Row 25; the first sentence may have a cited reference.

Response: We apologized for some missing references. We have checked the manuscript and added appropriate citations.

  1. Row 37; the space was necessary between Receptor and (MC4R). In the aspect of space, the overall text could be checked by the authors.

Response: Thanks for the reviewer’s suggestion, we have revised the relevant wording accordingly. Thank you for helping us improve the clarity of our paper.

  1. Row 193; resulte or result?

Response: Thanks for the reviewer’s suggestion, we have revised the relevant wording accordingly. Thank you for helping us improve the clarity of our paper.

Minor editing of English language required.

Response: Thanks for the reviewer’s suggestion, we have revised the relevant wording accordingly. Thank you for helping us improve the clarity of our paper.

Round 2

Reviewer 2 Report

Comments and Suggestions for Authors

The authors are thanked for addressing the previously raised issues, and for including more citations throughout the text and for using the citations in the tables as well Altogether, adding also new figures and a list of abbreviations, improved the structural overview and understandability of the topic.

It is is still suggested to do some proofreading and use spaces were new citations are added.

Author Response

The authors are thanked for addressing the previously raised issues, and for including more citations throughout the text and for using the citations in the tables as well Altogether, adding also new figures and a list of abbreviations, improved the structural overview and understandability of the topic. It is is still suggested to do some proofreading and use spaces were new citations are added.

Response: Thanks for the reviewer’s suggestion. We have carefully proofread the text, corrected any grammatical errors, and added spaces before citations. Thank you for helping us improve the clarity of our paper.

Reviewer 3 Report

Comments and Suggestions for Authors

The presence or absence of space before [reference No.] was not uniformed; for instance in row 45, the space was present before [7], while in row 47, the space was absent before [8-10].

Author Response

The presence or absence of space before [reference No.] was not uniformed; for instance in row 45, the space was present before [7], while in row 47, the space was absent before [8-10].

Response: Thanks for the reviewer’s suggestion. We have carefully proofread the text, and added spaces before citations. Thank you for helping us improve the clarity of our paper.